# *Anemarrhenae asphodeloides rhizoma* Extract Enriched in Mangiferin Protects PC12 Cells against a Neurotoxic Agent-3-Nitropropionic Acid

**DOI:** 10.3390/ijms21072510

**Published:** 2020-04-04

**Authors:** Agnieszka Piwowar, Nina Rembiałkowska, Anna Rorbach-Dolata, Arnold Garbiec, Sylwester Ślusarczyk, Agnieszka Dobosz, Anna Długosz, Zofia Marchewka, Adam Matkowski, Jolanta Saczko

**Affiliations:** 1Department of Toxicology, Wroclaw Medical University, 211 50556 Borowska, Poland; agnieszka.piwowar@umed.wroc.pl (A.P.); anna.rorbach-dolata@umed.wroc.pl (A.R.-D.); anna.dlugosz@umed.wroc.pl (A.D.); zofia.marchewka@umed.wroc.pl (Z.M.); 2Department of Molecular and Cellular Biology, Wroclaw Medical University, 211A 50556 Borowska, Poland; nina.rembialkowska@umed.wroc.pl (N.R.); jolanta.saczko@umed.wroc.pl (J.S.); 3Department of Developmental Biology, Institute of Experimental Biology, University of Wroclaw, ul. H. 21 50335 Sienkiewicza, Poland; arnold.garbiec@uwr.edu.pl; 4Department of Pharmaceutical Biology and Botany, Wroclaw Medical University, 211 50556 Borowska, Poland; sylwester.slusarczyk@umed.wroc.pl; 5Department of Basic Medical Sciences, Wroclaw Medical University, 211 50556 Borowska, Poland; agnieszka.dobosz@umed.wroc.pl

**Keywords:** *Anemarrhena asphodeloides*, mangiferin, 3-nitropropionic acid, cytotoxicity, PC12 cells

## Abstract

The rhizome of *Anemarrhena asphodeloides* Bunge, used in Traditional Chinese Medicine as a brain function-improving herb, is a promising source of neuroprotective substances. The aim of this study was to evaluate the protective action of xanthones from *A. asphodeloides* rhizomes on the PC12 cell line exposed to the neurotoxic agent—3-nitropropionic acid (3-NP). The xanthone-enriched fraction of the ethanolic extract of *A. asphodeloides* (abbreviated from now on as XF, for the Xanthone Fraction), rich in polyphenolic xanthone glycosides, in concentrations from 5 to 100 μg/mL, and 3-NP in concentrations from 2.5 to 15 mM, were examined. After 8, 16, 24, 48, and 72 h of exposure of cells to various combinations of 3-NP and XF, the MTT viability assay was performed and morphological changes were estimated by confocal fluorescence microscopy. The obtained results showed a significant increase in the number of cells surviving after treatment with XF with exposure to neurotoxic 3-NP and decreased morphological changes in PC12 cells in a dose and time dependent manner. The most effective protective action was observed when PC12 cells were pre-incubated with the XF. This effect may contribute to the traditional indications of this herb for neurological and cognitive complaints. However, a significant cytotoxicity observed at higher XF concentrations (over 10 µg/mL) and longer incubation time (48 h) requires caution in future research and thorough investigation into potential adverse effects.

## 1. Introduction

Currently, the worsening quality of the environment (e.g., pollution of air, water, and food) relates to increased exposure to different xenobiotics, which adversely affect human health, especially in industrialized countries [1]. Many exogenous agents are neurotoxic and influence the condition of the central nervous system (CNS) in different ways. The overproduction of reactive oxygen species (ROS) is indicated as one of the important inducers of tissue damage and the development of neurodegenerative disorders [2,3,4]. Neurotoxins are also indicated as a reason for the increased incidence of neurodegenerative diseases, such as Alzheimer’s, Parkinson’s, or Huntington’s. It is estimated that the incidence of dementia-related diseases in 2050 will have tripled since 1950 [5]. Unfortunately, despite an abundance of research, there are still no effective methods of prevention or treatment for these diseases. Natural products from plants attract attention, based on their widespread use in traditional and folk medicine for the treatment and prevention of different neurological disorders, including neurodegeneration [6,7].

One of the most well-known neurotoxic agents is 3-nitropropionic acid (3-NP), commonly used to induce neurotoxicity in both in vivo and in vitro studies [8]. The 3-NP naturally occurs in leguminous plants used to feed animals and is capable of poisoning grazing livestock. Moreover, it is produced by various fungi, including molds, such as *Aspergillus flavus-oryzae* group, which can produce on various foods as much as 200 mg per kg of substrate [9,10]. A source of accidental human poisoning may be a consumption of foodstuffs (sugar cane, cereals, etc.) infected with these molds. Even if the acceptable daily intake has been estimated as 25 μg/kg/day or 1.75 mg/day for a 70 kg human [10], it has been revealed that in humans exposed to low doses of 3-NP, acute encephalopathy, followed by dystonia, may develop [2,3,11]. Toxin-treated experimental animals demonstrated decreased motor performance, with degeneration found primarily in the striatum, but also in the hippocampus and thalamus. 3-NP action is particularly responsible for biochemical and morphological changes in cells. Neurotoxicity-related cell damage is often associated with oxidative stress caused by mitochondrial energy impairment, which in turn initiates a chain of pathophysiological processes leading to cell death. Brain cells are prone to oxidative damage because of a high rate of oxygen consumption [12]. Oxidative stress (OS) also plays an important role in the 3-NP-induced neurotoxicity. Moreover, 3-NP is a specific inhibitor of the mitochondrial complex II electron transport chain, so it consequently leads to an excessive ROS production, mitochondrial DNA damage, and loss of the physiological function of mitochondria. Hence, 3-NP induces cell death, via necrosis or apoptosis, especially these in the nervous system [13,14]. Some in vivo research has indicated that 3-NP acts on cortical activity, but its overall effect is complex, involving elements of depression and excitation, which is important in neurodegenerative pathogenesis. Therefore, 3-NP has been widely used as an experimental model of neurodegeneration as well as a model to study energy metabolism and cell death in vivo and in vitro [7,15,16,17].

Some natural compounds derived from plant extracts may prevent or decrease the damage caused by neurotoxic agents and seem to form a promising class of non-conventional protective agents against neurodegenerative disorders. Natural products isolated from such plants as *Ginkgo biloba*, *Centella asiatica,* or *Panax ginseng* are useful in protection against the development of such disorders, but the exact mechanisms of their actions are still not completely understood [18,19,20]. 

Among the several herbs that were traditionally credited with neurological and cognitive indications, some recent reports have revealed the protective effect of the rhizomes of *Anemarrhena asphodeloides* Bunge (Asparagaceae, subfamily Agavoideae) against glutamate–induced excitotoxicity and focal ischemic brain injury [21,22,23]. The rhizome of *A. asphodeloides* is a well-known and commonly used herbal drug in the traditional medicine of East Asia (Chinese, Japanese and Korean), under vernacular names—Zhimu, Yanghuzi, or Jino. The Traditional Chinese Medicine describes its bitter, sweet, and cold character; lung, stomach, and kidney meridians entered and mentions heat clearing, yin nourishing, and moisturizing action, which found indications for lung and stomach derived fevers, dry cough, night sweats, and menopausal syndromes, among others. It has been also applied in mixtures as balancing the unwanted heat generated by so-called warm herbs [24]. *Anemarrhenae rhizoma* has been also mentioned along with a few other Traditional Chinese Medicine (TCM) herbs (*Huperzia serrata*, *Rhodiola* sp., *Salvia miltiorrhiza*, *Panax ginseng*, *Pueraria lobata*) as a promising treatment and prevention option for senile dementia [25]. *Anemarrhenae asphodeloides rhizoma* is also listed in European Pharmacopoeia—monograph No. 2661E. The confirmed pharmacological activity of this herb includes antibacterial, antiviral, anti-inflammatory, antipyretic, anticoagulation, and antidiabetic [23,26,27]. The active constituents in *A. asphodeloides* belong to several phytochemical classes. Until now, more than 100 components have been isolated, such as: steroid saponins (total content in the rhizomes—about 6%), xanthones (2%), flavonoids, phenylpropanoids, alkaloids, anthraquinones, organic acids, and others. The summary of phytochemistry and phytopharmacology of this herb has been provided by two comprehensive reviews [23,28]. As quality and activity markers, xanthone glycosides (mangiferin, isomangiferin, neomangiferin—Figure 1) and saponins (several timosaponins with the sarsapogenin as aglycon) are mentioned.

The existing information about this TCM herb prompted us to examine the potential protective action of *A. asphodeloides* towards toxicity induced by the neurotoxic agent-3-NP, especially with regard to the limited amount of information concerning polyphenols contained in the polar extract. A model for this study, the PC12 cell line, derived from a pheochromocytoma of the rat adrenal medulla, is often used in evaluation of neurotoxicity and neuroprotective effects of different compounds, particularly after Nerve Growth Factor (NGF)-induced differentiation. The PC12 line is suitable for neurotoxicity (especially excitotoxicity) evaluation, due to the quality and stability of its enzymatic and metabolic background [29,30]. The aim of this study was to examine the protective action of xanthone-enriched fraction (XF) of the ethanolic extract from the rhizomes of *A. asphodeloides*, in the model of the PC12 cell line, exposed to the well-known and commonly used neurotoxic agent-3-nitropropionic acid. We examined different combinations of each compound: ethanolic extract enriched in xanthone glycosides of *A. asphodeloides* (XF) and 3-NP, in different concentrations, times, and sequences of experiments, which, to our best knowledge, has not been done so far.

## 2. Results

### 2.1. Effects of 3-NP on the PC12 Cells Viability 

In the single factor cytotoxicity assay, a dose-dependent decrease of PC12 cell viability was observed upon 3-NP addition (Figure 2A). These results indicated that treatment for 48 h reduced cell viability by approximately 50% at 2.5 mM 3-NP treatment, and treatment for 24 h reduced cell viability by roughly 50% at 10 mM 3-NP, while treatment for 8 h reduced cell viability by approximately 50% at 15 mM 3-NP. Therefore, we used various concentrations of 3-NP (2.5, 5, 10, and 15 mM) to treat PC12 cells during different time in the subsequent experiments and compared the cell survival percentage to the values of single-treatment with 3-NP.

### 2.2. Effects of XF on PC12 Cells Viability 

The results of the single-factor treatment with the xanthone fraction showed low toxicity toxic effects on PC12 cells only for the five lowest concentrations (up to 15 µg/mL) during first 24 h. During these two periods of treatment (8 and 24 h) the linear dose-response was observed (regression analysis: 8 h, *y* = −0.840334*x* + 100.57533, *r*^2^ = 0.9869, 24 h, *y* = −0.812092*x* + 103.04335, *r*^2^ = 0.9641) with less than 50% live cells remaining at 70 µg/mL. The 48h-long treatment caused non-linear (regression function as *y* = 104.017526 × e^ (−5.664033769 × 10^−2^*x*), significant at *p* < 0.005) response to increased dose with a dramatic drop in the cell survival over 5 µg/mL. These results were used as a clue for the concentration range used in further experiments (Figure 2B).

### 2.3. Effect of XF on the PC12 Cells under 3-NP-Induced Cytotoxicity

PC12 cells were treated with higher concentrations of XF (5, 10, and 15 µg/mL), interchangeably with 3-NP (2.5, 5, 10 mM) for an incubation time of 8 or 48 h. The first set of experiment included pre-incubation with either XF or 3-NP, followed by the addition of the other compound (8 h total incubation time in Figure 3A and 48 h total incubation time in Figure 3B).

For the shorter incubation (8 h), the addition of the XF prior to 3-NP (Figure 3A, white bars) for the first 4 h resulted in statistically insignificant or weakly significant (*p* > 0.05) increase of cell survival in comparison to the 3-NP treated cells with no clear relationship to the applied XF concentration. Conversely, adding 3-NP first, followed by XF after 4 h (Figure 3A, dark grey bars), resulted in further increased cytotoxicity at 15 µg/mL XF combined with 2.5 or 5.0 mM 3-NP (82.2% vs. 89.9% and 61.1% vs. 78.4% surviving cells, respectively; significant at *p* < 0.005). 

For the longer incubation (48 h), the addition of the XF prior to 3-NP (Figure 3B, white bars) the cell survival was either insignificantly (at 2.5 mM 3-NP vs. 5 µg/mL XF and 5 mM 3-NP vs. all three XF concentrations), or significantly (at *p* < 0.005 in all other combinations) reduced compared to the single 3-NP treatment. Therefore, an additive cytotoxic effect may exist in higher concentrations and after the longer incubation time.

On the contrary, the addition of XF after 24 h challenge with 3-NP (Figure 3B), dark grey bars) resulted in lack of any effect on cell viability (*p* > 0.05).

In the subsequent set of experiments, the PC12 cells were treated with very low concentrations of XF (0.5 and 1 µg/mL), interchangeably with 3-NP (2.5, 5, 10, 15 mM) for 48 h. 

Contrary to the higher XF concentrations, a pronounced protective effect was observed in both ways of treatment, i.e., in 3-NP and XF preincubation (Figure 3C,D, respectively). When 3-NP was added first and XF after 24 h, the full alleviation of 3-NP toxicity was noticed at 2.5 mM. At 5.0 mM 3-NP, only the higher (1 µg/mL) concentration of XF caused a return of cell viability to 100%. Even at the higher 3-NP doses (10 or 15 mM), there was a significant increase in cell viability (*p* < 0.05 for 10 mM 3-NP + 0.5 µg/mL Xf and *p* < 0.005 for other combinations). This effect was even stronger when the cells were preincubated with respective doses of XF, and 3-NP was added after 24 h (Figure 3D). Only at the 3-NP concentration of 15 mM, the live cell percentage did not return to 100%, and was 93.3% ± 54% to 94% ± 4.9% for 0.5 µg/mL and 1.0 µg/mL of the xanthone fraction, respectively.

### 2.4. Effects of Combined XF Pre-Incubation and Co-Incubation XF 

PC12 cells were treated with 3-NP (2.5, 5, 10, and 15 mM) with the absence or presence of low concentrations of XF (0.5 and 1 µg/mL) for 8 h and 24 h, after pre-incubation of the cells with XF in concentrations of 0.5 and 1 µg/mL in cell culture flasks for at least 48 h.

A very contrasting results were observed between 8 h and 24 h incubation with 3-NP (Figure 3 A vs. B). After 8 h incubation with 3-NP (Figure 3A), a highly significant (in most combinations but two) alleviation of cytotoxicity occurred. At 2.5 mM 3-NP, the effect was fully abolished in 0.5 µg/mL pre-incubation (106.2% ± 2.2% of cells survived, compared to the untreated). Under additional co-incubation with 0.5 µg/mL the effect was weaker (98.3% ± 5.5%) but also highly significant (*p* < 0.005). Interestingly, at the higher concentration of XF (1 µg/mL), the positive effect on cell survival was only in pre-incubated samples, while in the co-incubated samples, the effect was noticeable (94.1% ± 7.2% vs. 89.8% ± 3.7%) but statistically insignificant (*p* > 0.05). 

At 5.0 and 10.0 mM 3-NP, the effect was highly significant in all combinations with similar tendencies—pre-incubation with just 0.5 µg/L gave the highest level of cell viability (Figure 4A, black bars), followed by pre- and co-incubations with 0.5 µg/mL XF (Figure 4A, dark grey bars). 

At the highest used 3-NP concentration (15 mM), the trends were rather different. Here, the combined pre- and co-incubation were highly significantly protecting the cells at both 0.5 and 1.0 µg/mL. Here, the pre-incubation only experiments failed to prove such a marked protecting effect having only weakly significant (*p <* 0.05) at 0.5 µg/mL XF or no effect at all (at 1.0 µg/mL).

The 24h incubation resulted in decrease cell survival upon pre-incubation with XF, disregarding of the XF concentration (highly significant at *p* < 0.005). This detrimental (co-toxicity) effect was only partly alleviated upon co-incubation with an additional XF dose (Figure 4B) that lead to the cell survival rate set back to the level of 3-NP only treatment.

### 2.5. Confocal Scanning Laser Microscopy for Detecting the Evaluation and Intracellular Distribution of Mitochondria

PC12 cells were treated with various concentrations of XF (0.5, 1, 5, 10, 15, and 20 µg/mL), interchangeably with 3-NP (2.5, 5, 10, 15 mM), for 24 h. Control cells, cultured in RPMI with no additional treatments had regular round nuclei, chromatin distributed evenly with prominent nucleoli and uniformly dispersed fluorescence derived from mitochondria. All cells were similar in size and shape. After incubation with the XF at the concentration of 10 µg/mL, significant increases in the cumulative and fluorescence intensity of mitochondria were observed (Figure 5). After incubation with the XF at the concentration of 15 µg/mL, the cells manifested different morphology and sizes characterized by fragmentation of the nucleus, an excessive condensation of chromatin as well as a weak signal from MitoTracker^®^ Deep Red. It was observed that after incubation with the XF at concentrations of 5 µg/mL and 5 mM NP-3, the cell nuclei are significantly shrunk, however the signal from the mitochondria was normal. There were no cells with disintegrating nuclei or mitochondria (intense point fluorescence) after incubation with XF at concentrations of 10 µg/mL and 5 mM 3-NP. After incubation with XF at 15 µg/mL, undulating nucleus membranes were observed. This demonstrates that incubation with the XF at this concentration had no effect on preventing the effects of 3-NP.

PC12 cells after 24 h of pre-incubation of XF at concentrations of 0.5 and 1.0 µg/mL did not exhibit significant changes in the shape of the nuclei in comparison to the controls (Figure 6). However, chromatin condensation was perceived in the nucleus of individual cells. PC12 cells pre-treated with low concentrations of XF demonstrated higher red mitochondrial signal intensity than the control cells. In both cases, single vacuoles were observed and the fluorescence signal from the mitochondria formed a large point. After 24 h, incubation of the cells with 2.5 mM 3-NP, normal morphology, cells with a wrinkled nucleus and nuclei with a few vacuoles were observed. PC12 cells which were pre-incubated with XF at a concentration of 0.5 µg/mL and with 2.5 mM 3-NP were characterized by a variety of cell shapes and sizes and irregular nuclei. Invagination of the cell membrane was observed in cells pre-incubated with XF at both concentrations of 0.5 and 1.0 µg/mL after incubation with 2.5 mM 3-NP. The cells after incubation with 10 mM 3-NP were characterized by a strong corrugated nucleus. The morphology of cell nuclei after pre-incubation with XF at concentrations of 0.5 and 1.0 µg/mL XF and incubation with 10 mM 3-NP was correct, but a few highly corrugated cell nuclei were also perceived. The cells after incubation only with 15 mM 3-NP suffered from a degenerative nucleus. The cells after incubation with XF at concentrations of 0.5 and 1.0 µg/mL and incubation with 15 mM 3-NP were not as degenerated as cells incubated only with 15 mM 3-NP. All the cells described above after incubation with 15 mM solution of 3-NP were characterized by numerous vacuoles.

### 2.6. Determination of Polyhydroxyxanthone Glycosides

The content of mangiferin in the tested extract was 293.72 ± 6.63 mg/g (on dry mass basis) and the total xanthones content was 35.45% (mangiferin plus two conspicuous peaks with matching UV spectra, tentatively identified as neomangiferin—37.90 ± 0.24 mg/g, isomangiferin 22.91 ± 0.11 mg/g). None of the other constituents (peaks) detected between 210 and 500 nm exceeded 2% of the total chromatogram peak area. Therefore, the extract can be considered as rich in polyphenolic xanthone glycosides, with mangiferin making up the largest proportion (Figure 7). 

## 3. Discussion

The PC12 cell line is a widely used experimental model for neuroprotection and neurodegeneration analysis, particularly in the contexts of excitotoxicity [30,31], whereas 3-nitropropionic acid is usually applied as a neurotoxic agent trigger for mitochondrial dysfunction [13]. We used both in our in vitro study for the examination of the potential protective action of the ethanolic extract of *A. asphodeloides* against neurotoxic agents. We examined different combinations of both these compounds, XF and 3-NP, in different concentrations, times, and sequences of experiments. These allowed us not only to examine the potential protective action of XF, but also to estimate its possible adverse effects. Additionally, to the best of our knowledge, the combination of pre-incubation, co-incubation and a mixture of both with plant polyphenols upon 3-NP treatment, has not previously been examined in this particular model.

Nonetheless, the striking discrepancies between the low and higher dosage treatments and quite complex relationships with the toxic agent and experiment timing call for cautious conclusions about the potential health benefits. Moreover, before initiating any in vivo experiments, the adequate dosage scheme should be meticulously established. 

The cytotoxicity of the *A. asphodeloides* xanthone fraction was observed already at the relatively low concentrations. However, the conspicuous cytotoxic effect occurred only after a long (48 h) incubation. It would suggest that it was rather a secondary effect, most probably related to the complex redox physiology than direct damaging properties of polyphenols. Among the feasible mechanisms we suspect prooxidant activity resulting from formation of free-radical intermediates (such as aromatic radical cations) and/or interactions with transition metal ions, such as copper. The increasing oxidative molecules not only damage cell components directly but also release a cascade of responses in such organelles as ER or mitochondria [12,32]. The latter mechanism has been postulated as a major cause of cytotoxic properties of higher doses of polyphenols, useful for anticancer activity [32] but potentially harmful when cells must be protected (in neurodegeneration) [33]. Our microscopic observations show the mitochondrial damage even in cells incubated with the xanthone fraction only. Thus, mitochondria are at least one of the target structures that suffer from excessive doses and prolonged exposition to *A. asphodeloides* xanthone fraction. This phenomenon could be also exploited for further anticancer studies as it has been already demonstrated for other constituents from *A. asphodeloides rhizoma,* i.e., timosaponin A-III and other triterpenoids [34,35]. 

Furthermore, we observed an apparent co-toxicity with 3-NP. The 3-NP is an oxidative stress- generating agent that targets neuronal mitochondria, so the accumulation of prooxidant species from such highly hydroxylated aromatic molecules as xanthones would be a plausible explanation for this [36]. However, a thorough metabolomic investigation will have to be performed to verify a presence of such hypothetical prooxidant xanthone intermediates in the challenged cells. 

Even so, there are quite many other studies that show beneficial activities of plant derived natural products in alleviation of oxidative stress related cellular damage [37]. Among the compounds showing such properties, there are many polyphenols, such as resveratrol [12], anthocyanins [38], caffeic acid oligomers [39], and many other [37]. Hydroxylated xanthones have been also reported as able to protect against neuronal damage in various models. α-Mangostin, a prenylated non-glycosylated xanthone, reduced oxidative damage in rat brain tissue [40] and primary cultures of cerebellar granule neurons pre-incubated for 30 min with α-mangostin (0.82–8.21 µg/mL) and treated with 3-NP at 2 mM for another 24 h [41]. These authors report a significant cytoprotective effect of α-mangostin at concentrations of 2.46 µg/mL or more (up to 4.94 µg/mL) with a cytotoxic effect observed at 6.56 µg/mL. These results corroborate with our data regarding cytotoxicity but cytoprotective effect of XF occurred at much lower concentrations. However, in the cited study, no other combinations were tested. Mangiferin, the major component of our XF fraction was tested in a recent study by Peng et al. [42] using PC12 cells damaged with oxidative agents (H_2_O_2_ and 6-hydroxydopamine). The cytoprotective effect measured, in addition to MTT, by lactate dehydrogenase (LDH) release and caspases activity was significant at 8.45 µg/mL and 21.12 µg/mL, but the full restoration to control levels had not been reached. The authors did not report any cytotoxicity. However, also in this report, only one combination was used, i.e., 24 h pre-treatment with mangiferin followed by 12 h with oxidizing agents. Mangiferin treatment caused upregulation of endogenous antioxidant system via increase of antioxidant response element (ARE) driven genes and the essential role of Nrf2 was illustrated by increased translocation of this protein to the nucleus. The proof for Nrf2 activation as a key mechanism of cytoprotection was provided by using knockdown cells with inactive Nrf2 pathway. It is highly probable that, also, *A. asphodeloides* xanthone fraction exerts its effect via preventing Nrf2 degradation by mangiferin and other hydroxyxanthones. Saponins, the other major phytochemical class contained in the rhizomes demonstrate neuroprotective effects on cerebral ischemia injures in the animal model [43,44]. Three other minor phenolic compounds: 2,6,4’-trihydroxy-4-methoxybenzophenone, 7-hydroxy-3-(4-hydroxybenzyl)-chroman, and broussonin B isolated from *A. asphodeloides rhizoma* stimulated neurite outgrowth in PC-12 cells [45]. However, none of these individual components have been pinpointed as sole determinants of the traditional usage for treatment of CNS disturbances. Most probably, the complex action of compounds from various phytochemical classes may result in a beneficial net effect. However, if the mechanism of action involves antioxidant properties, xanthones are the major active antioxidant to recommend [38,40]. 

A few published in vivo studies support the function of xanthones as neuroprotective agents acting via antioxidant mechanisms. Xi et al. [46] fed rats with mangiferin prior to insult with cerebral ischemia. Mangiferin administration (50–200 mg/kg bw) reduced infarct areas dose dependently. The effect was potentiated by isoflurane treatment leading to the conclusion that this drug combination acted synergistically via inhibition of ROS-mediated apoptosis and activation of phsphatidylinositol-3-kinase/Akt/mTOR pathway. 

In mice with severe memory deficits associated with elevated ROS, malondialdehyde (MDA) levels and acetylcholinesterase (AChE) activity, attenuated superoxide dismutase (SOD), catalase (CAT) and glutathione (GSH) activity, showed that treatment with mangiferin improved cognitive performance by the significant reduction of prooxidative parameters and elevated antioxidant defence parameters [47]. Zajac et al. [48] revealed that mangiferin derived from the mango tree bark (*Mangifera indica* L.) can improve brain functions, but it is unlikely that the compound traverses the blood-brain barrier after being systemically administered in rats. The authors suggest that it is improbable that mangiferin could act via direct interaction with central neural components, but rather has peripheral, target specific functions, which could be secondarily reflected in the brain metabolism. However, Peng et al. [42] argue that despite theoretical limits for an intact xanthone glycoside to cross the blood-brain barrier (BBB), the site-specific opening of this barrier might facilitate the action directly in the brain tissue. Here, we can also speculate that the crude extracts as used traditionally could benefit from saponin presence not only as active compounds but also as adjuvants facilitating targeted distribution of other compounds such as polyphenols via vesicular transport [49]. 

Literature data indicate that protective activity is also determined by the method of processing of the plant and/or extraction of the individual compounds. In the case of *A. asphodeloides* it has been proven that the process using mixing-bake enhances the hypoglycemic effect but reduces the anti-inflammatory effect. Different methods of processing the rhizome are responsible for varying degrees of antibacterial activity. On the other hand, the aqueous extract of *A. asphodeloides* exhibits greater neuroprotective activity towards damage due to focal cerebral ischemia [21,23]. The mechanisms of change in pharmacological effects are not explained in detail. However, it has been shown that there are no differences in the chemical composition of the components between the various methods of treating *A. asphodeloides* rhizome. It seems that various pharmacological effects are not the result of the elimination of selected constituents but may be caused by changes in the quantity of bioactive ingredients [50]. In our study, we used ethanol extract of rhizome obtained by the recommended pharmacopoeia method (further enriched in mangiferin), and significant protective properties were observed along with exacerbated toxic effects in higher doses. This indicates a need for further studies on the mechanisms of action and pharmacological effects of *A. asphodeloides* constituents, as well as looking for confirmation of their safety and effectiveness as a potential therapeutic agent. Further research will be carried out, due to the limited data on the neuroprotective properties of XF, and we plan to incubate PC12 cells differentiated into neuronal cells with NGF factor, with XF concentrations selected based on the results of the current study, and investigate the mechanism including its neuroprotective action. This plant has acclimatized very well to the conditions in Central Europe, and its cultivation and the acquisition of raw materials for production would be both technically possible and economically attractive. 

## 4. Materials and Methods 

### 4.1. Preparation of Anemarrhena Asphodeloides Extract 

#### 4.1.1. Isolation Procedure 

*Anemarrhena asphodeloides* rhizomes were harvested in the fall of 2015 from plants cultivated in the certified collection of the Botanical Garden of Medicinal Plants, Wroclaw Medical University (Wroclaw, Poland, geographic coordinates 51°07’03’’N, 17°04’27’’E, altitude 117 m a.s.l.). The voucher specimen is stored in the Botanical Garden herbarium under the code “2015-Anemarrhenaasphodeloides-1-FBBO”. The root powder was prepared in accordance with the recommendations of the Polish Pharmacopoeia (IX edition). The drug quality was ascertained according to the European Pharmacopoeia monograph 2661E. The xanthone fraction (XF) of *A. asphodeloides* rhizomes was prepared using the procedure described below.

Powdered plant material (200 g) was defatted by maceration in 1.5 L *n*-hexane (analytical grade, Chempur, Gliwice, Poland) in a flat-bottomed flask for 2 h on an orbital shaker (Elpin Plus, Lubawa, Poland) at 100 rpm. Then, the hexane was removed by filtering and the rhizome powder was thoroughly dried under a fume hood, mixed with 1 L of 80% aqueous ethanol and sonicated (IS-33 ultrasound bath, Intersonic, Olsztyn, Poland) for 2 h. Then, the extract was filtered through analytical filter paper (Equimed, Krakow, Poland). The crude extract was passed through octadecyl-silyl ODS silica gel (50 g, Davisil 130, Grace, Columbia, MD, USA) in a 500 mL glass Büchner funnel (Schott, Mainz, Germany), washed with purified (HPLC grade) water (eluate discarded), followed by 40% ethanol, and evaporated in a rotary vacuum evaporator at 40 °C until a yellow-orange, dry, crystalline powder was obtained. The extraction efficiency was 25 g from 200 g of powdered plant material. Further enrichment procedures were as previously published [51]. The dried crystalline extract was powdered using a mortar and pestle and dissolved in 80% rectified, food grade ethanol (*v/v* in water, Polmos Wroclaw, Poland). The extract solution was sterilized by a syringe filter with a pore size of 0.22 μm. The extract, dissolved in 80% ethanol, was used in all further experiments. XF was added to the PC12 cell culture plates in different combinations and incubated in an appropriate manner (described in detail in Section 4.5.). 

#### 4.1.2. Analysis of Extract Composition 

The presence of xanthone glycosides and saponins was confirmed with a thin layer of chromatography (TLC), according to the European Pharmacopoeia Monograph #2661 on the *Anemarrhena asphodeloides* rhizome. The content of mangiferin and total xanthone derivatives was measured using a reversed phase HPLC method adapted from European Pharmacopeia. Briefly, the Merck Hitachi (Hitachi Ltd., Tokyo, Japan) LaChrom 7000 series system with quaternary low- pressure gradient pump (L-7100), in-line degasser (L-7641), autosampler, and diode array detector (L-7450). The analytical column Cadenza CD C18 3 μm 150 mm × 4 mm (Imtakt, Portland, OR, USA) was used for separation. Solvent consisted of acidified aqueous acetonitrile (ACN: 0.2% acetic acid 15: 85 *v/v*). Isocratic run was used at a flow rate of 1mL/min, run time 30 min, injection volume 10 μL. Detection wavelength was 258 nm and the identification of xanthone peaks was based on retention time comparison with standard mangiferin (analytical standard, Sigma-Aldrich, St. Louis, MO, USA) and characteristic UV spectra match. Quantification of total xanthones was based on the mangiferin calibration curve, using peak area.

### 4.2. Cell Culture

The PC12 cell line derived from pheochromocytoma of the rat adrenal medulla was cultured in the RPMI 1640 medium (R8758-500 mL, Sigma, St. Louis, MO, USA) with 10% fetal bovine serum (FBS) (Sigma, USA), supplemented with 1% penicillin–streptomycin (Sigma, USA). The cell line was derived from the European Collection of Authenticated Cell Cultures (ECACC). Cells were cultured in plastic flasks, 25 or 75 cm^2^ (NalgeNunc, Rochester, NY, USA), and incubated at 37 °C in a humidified atmosphere containing 5% CO_2_ in an incubator (SteriCult, ThermoScientific, Warsaw, Poland). The cell line was harvested during the exponential growth chase using 0.02% Trypsin EDTA (Sigma, USA) [52].

### 4.3. Cell Viability Assay 

Cell viability was determined by MTT (3-(4,5-dimethylthiazol-2-yl)-2,5-diphenyltetrazolium bromide) tetrazolium reduction assay that assesses the reduction of MTT to formazan by living cells. MTT was measured according to the manufacturer’s protocol. This is a colorimetric assay for assessing cell metabolic activity (the number of viable cells) based on NAD(P)H-dependent cellular oxidoreductase enzymes connected with mitochondrial membranes. These enzymes can reduce the tetrazolium dye MTT to its insoluble formazan, which has a purple color. After the conduction of appropriate experiments, the medium of each well was replaced with 10 μL of 0.5 mg/mL MTT stock solution (; In Vitro Toxicology Assay, Sigma St. Louis, MO, USA) diluted in 90 μL PBS. After 2 h of incubation, isopropanol with 0.04 M HCl was added (100 μL/well). The absorbance was quantified using a multiwell scanning spectrophotometer at 570 nm (Multiscan MS microplate reader, Artisan Technology Group, Champaign, IL, USA). Cell viability (reflected by mitochondrial activity), in comparison to the control group including viable nontreated cells, was expressed as a percentage value [52,53]. 

### 4.4. 3-NP-Induced Cytotoxicity in PC12 Cells

To induce cytotoxicity, PC12 cells were treated in a 96-well plate for 8, 24, or 48 h by adding various concentrations (from 0 to 15 mM) of 3-NP as a neurotoxic agent (conditions described in Section 4.2). After an appropriate time, the medium was removed and the MTT assay was performed to assess cell viability. For estimation of the potential protective effect of *A. asphodeloides*, the different combinations of XF and 3-NP were examined (described in detail in Section 4.5) [54].

### 4.5. Evaluation of Protective Activity against 3-NP–Induced Cytotoxicity in PC12 Cells

#### 4.5.1. Experimental Models 

For examination of the potential protective activity of XF, experiments were conducted in two ways. In the first way: the cells were detached and seeded on 96-well plates at a density of 300,000 cells/mL, and 24 h after seeding, the cells were treated with 3-NP (at a concentration from 0 to 15 mM). After the next 24 h, 3-NP was removed and different concentrations of XF (from 0.5 to 15 μg/mL) were added and examined for an additional 24 h. The second way was as follows: the PC12 cells were pre-incubated for 24 h of with different concentrations of followed by additional 24 h incubation with 3-NP. In that manner, PC12 cells were cultured in RPMI 1640 with 10% FBS, 1% penicillin–streptomycin and the XF fraction. The XF concentrations were 0.5 and 1 μg/mL. The cells growing in culture vessels were detached and seeded on 96-well plates at a density of 6 × 10^4^ cells/well with 200 μL RPMI 1640 containing 10% FBS, 1% penicillin–streptomycin, with/without the XF (0.5 or 1 μg/mL) in a 96-well plate and were incubated at 37 °C. 24 h after seeding, the cells were treated with 3-NP (at a concentration from 0 to 15 mM) with the absence or presence of different concentrations of XF (from 0.5 to 15 μg/mL) for 8 and 24 h. All applied combinations are presented in detail in the Results section (Figure 2, Figure 3 and Figure 4).

#### 4.5.2. Confocal Scanning Laser Microscopy Study 

PC12 cells 24 h after incubation with XF and/or 3-NP were incubated for 45 min with 250 nM Mitotracker^®^ Deep Red (ThermoFisher Scientific, Waltham, MA, USA), that stains mitochondria in live cells and can be used for mitochondrial localization, and 4′,6- diamidino-2-phenylindole (DAPI) (Sigma-Aldrich, St. Louis, MO, USA) used to detect the nucleus. Cells were visualized under a confocal microscope, Olympus (Tokyo, Japan) Fluoview 1000 UPlanSApo (lens magnification 60×), and the Mitotracker Deep Red fluorescence was determined at an excitation wavelength of 635 nm and 655–755 nm. DAPI fluorescence was determined at an excitation wavelength of 405 nm, and emission was detected at 433–455 nm. 

### 4.6. Statistical Analysis

Data were analyzed statistically using Microsoft Office Excel 2016. Each outcome is a mean of results measured for at least four samples. The results of the MTT assays were reported as means ± SD. To evaluate distribution of the results, the Kolmogorov–Smirnov test was used. Statistical significance was determined by Student’s *t*-test for independent samples: *p* < 0.05 or *p* < 0.005 values (two-tailed test) were assumed as statistically significant. The *t*-test was used for means comparison between treatments and the negative (vehicle only) samples. The XF samples were also compared to corresponding 3-NP treatments to reveal statistical significance of the observed outcome (set as *p* < 0.05 or 0.005). For dose-dependency of single 3-NP or XF treatments, we used linear or non-linear regression fit (least squares approach) of GraphPad Prism v. 7 (GraphPad Software, San Diego, CA, USA).

## 5. Conclusions

Our study revealed that the xanthone-enriched fraction of ethanolic *Anemarrhena asphodeloides* rhizome extract enhances cell proliferation and curbs the cytotoxic activity of 3-NP. However, the effect depended on the concentration, incubation time and the way cells were treated with the plant extract. The strongest protection was observed for pre-incubated cells at a very low concentration of 0.5 μg/mL, which seems very promising with respect to its use preventively in the development of neurodegenerative diseases. The addition of the xanthone fraction to a culture successfully prevented changes in cell morphology and apoptosis caused by 3-NP, as confirmed by cell imaging with the use of a confocal microscope. On the other hand, the significant cytotoxicity of the XF at higher concentrations and upon longer incubation as well as exacerbation of 3-NP toxicity after 48 h requires deeper insight into the cellular and molecular mechanisms of metabolism of the bioactive phytochemicals.

A further, deeper study focused on the neuroprotective action of *Anemarrhenae asphodeloides rhizoma* is required, as the results of our work are promising but not without controversy.

## Figures and Tables

**Figure 1 ijms-21-02510-f001:**
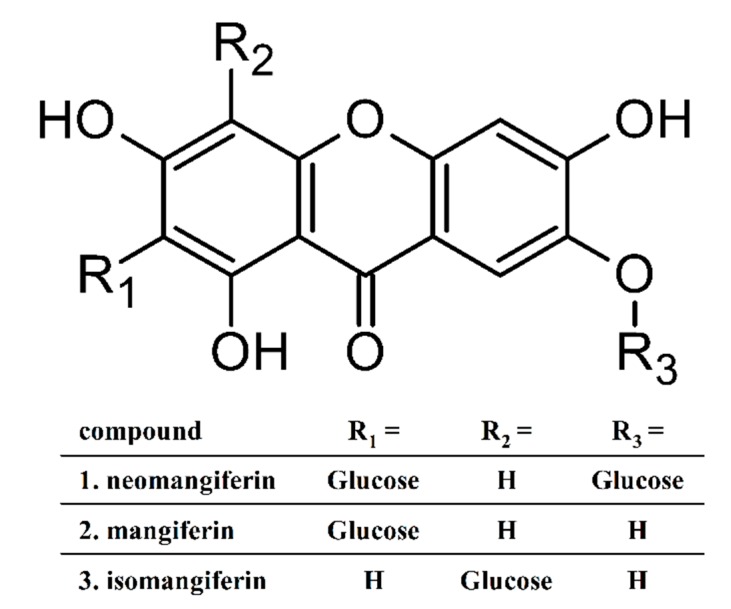
The structure of mangiferin (2) and the two minor xanthone C-glucosides—neomangiferin (1) and isomangiferin (3). The numbering corresponds to the peak elution sequence in the HPLC shown on Figure 7.

**Figure 2 ijms-21-02510-f002:**
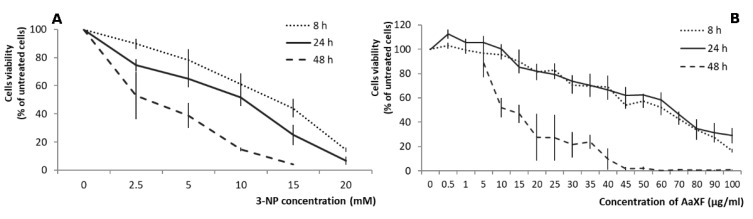
(**A**). Effects of 3-nitropropionic acid (3-NP) on cell viability in PC12 cells. PC12 cells were treated with various concentrations of 3-NP (0, 2.5, 5, 10, 15, and 20 mM) for various times (8 h, 24 h, and 48 h). (**B**) Effects of xanthone-enriched fraction (XF) on cell viability in PC12 cells. PC12 cells were treated with various concentrations of XF (from 0.5 to 100 µg/mL) for various times (8 h, 24 h, and 48 h). The cell viability was measured using the 3-(4,5-dimethylthiazol-2-yl)-2,5-diphenyl tetrazolium bromide (MTT) assay. The data represent the mean ± SD of reduction in cell viability of three independent experiments.

**Figure 3 ijms-21-02510-f003:**
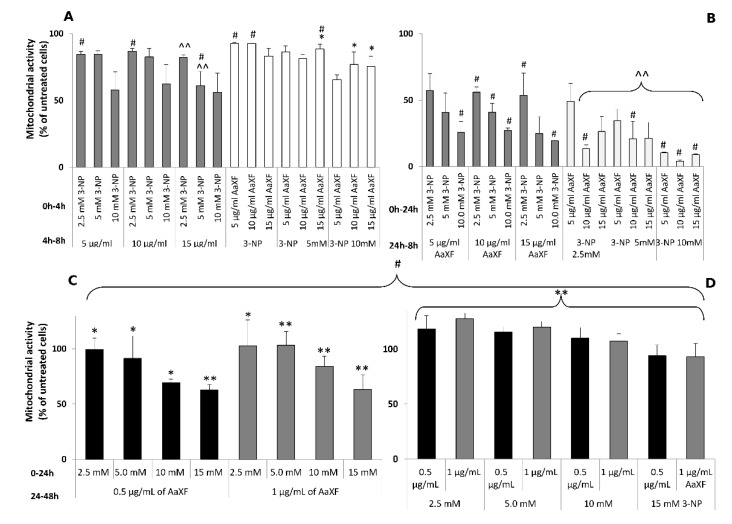
Mitochondrial activity of PC12 cells which were treated with various concentrations of XF (5, 10, and 15 µg/mL), interchangeably with 3-NP (2.5, 5, 10 mM) for 8 h (**A**) and 48 h (**B**), grey bars represent pre-incubation with 3-NP and the white bars represent pre-incubation with the XF. Mitochondrial activity of PC12 cells which were treated with very low concentrations of XF (0.5 and 1µg/mL), interchangeably with 3-NP (2.5, 5, 10, 15 mM) for 48 h ((**C**) pre-incubation with3-NP, (**D**) pre-incubation with the XF). The mitochondrial activity of cells was measured using the 3-(4,5-dimethylthiazol-2-yl)-2,5-diphenyl tetrazolium bromide (MTT) assay. Data represent the mean ± SD of reduction in cell viability of five independent experiments. Significantly higher from the corresponding 3-NP treatments group at * *p* < 0.05 or ** *p* < 0.005 and significantly lower from the corresponding 3-NP treatments at ^ *p* < 0.05 or ^^ *p* < 0.005. # indicates statistically significant (*p* < 0.05) difference between means of 3-NP pre-incubation vs. XF pre-incubation at the same concentrations.

**Figure 4 ijms-21-02510-f004:**
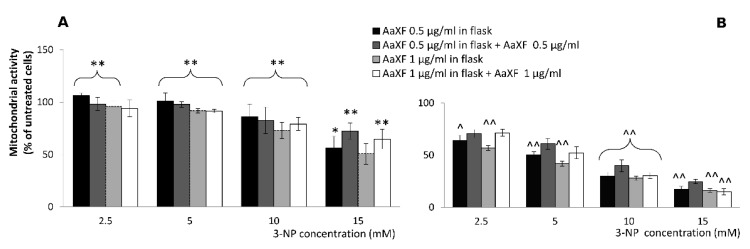
Mitochondrial activity of PC12 cells after pre-incubation of XF (0.5 or 1 µg/mL) in cell culture flasks for at least 48 h, which were treated with 3-NP (2.5, 5, 10 and 15 mM) with the absence or presence of very low concentrations of XF (0.5 and 1 µg/mL) for 8 h (**A**) and 24 h (**B**). The mitochondrial activity of cells was measured using a 3-(4,5-dimethylthiazol-2-yl)-2,5-diphenyl tetrazolium bromide (MTT) assay. The data represent the mean ± SD of cell viability in five independent experiments. Significantly higher from the corresponding 3-NP treatments group at * *p* < 0.05 or ** *p* < 0.005 and significantly lower from the corresponding 3-NP treatments at ^ *p* < 0.05 or ^^ *p* < 0.005.

**Figure 5 ijms-21-02510-f005:**
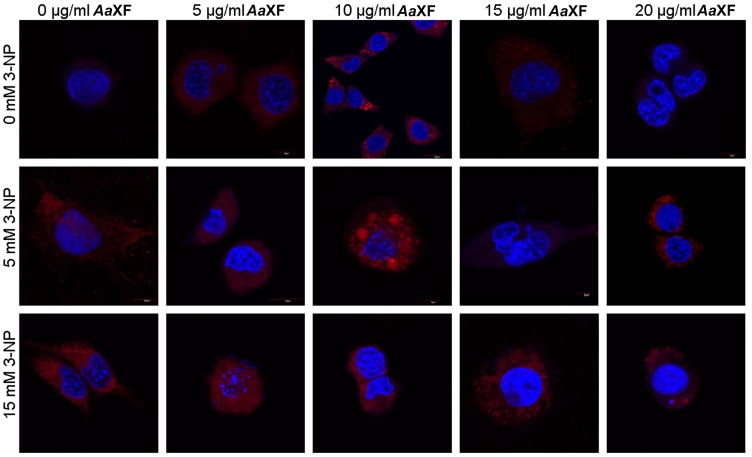
Confocal fluorescence analysis of PC12 cells using MitoTracker^®^ Deep Red FM and 4′,6- diamidino-2-phenylindole (DAPI) after incubation with the xanthone fraction of *Anemarrhena asphodeloides* (5, 10, 15, and 20 µg/mL) and 3-NP (5, 15 mM). Scale bar = 10 µm

**Figure 6 ijms-21-02510-f006:**
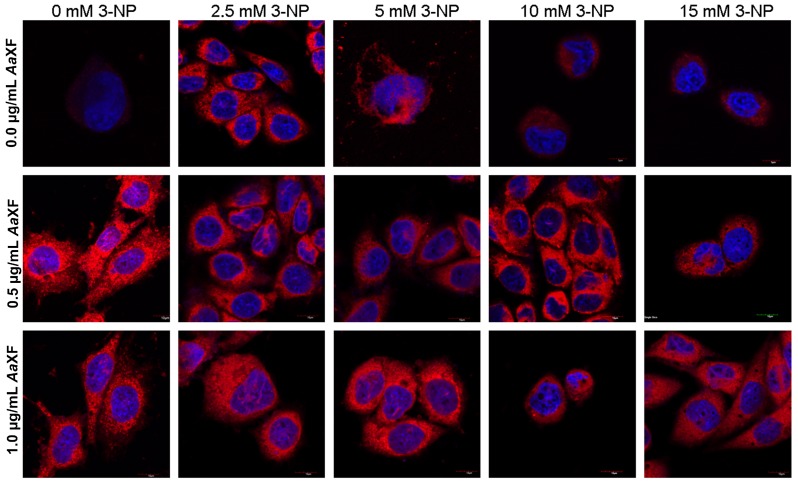
Confocal fluorescence analysis using MitoTracker^®^ Deep Red FM and DAPI of PC12 cells cultured in RPMI with added plant extract (pre-incubation) followed by incubation with the xanthone fraction of *Anemarrhena asphodeloides* (0.5, 1 µg/mL) and 3-NP (2.5, 5, 10 and 15 mM). Scale bar = 10 µm.

**Figure 7 ijms-21-02510-f007:**
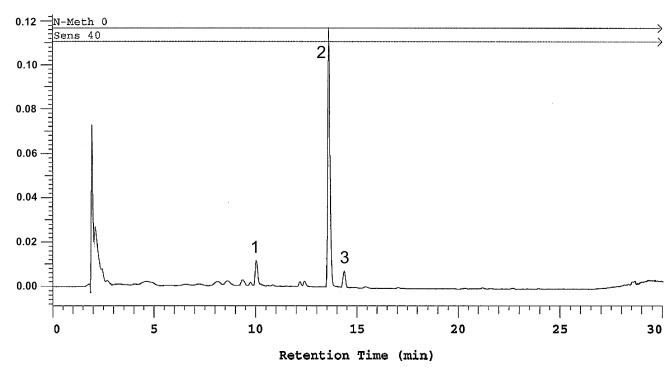
Results of RP-HPLC analysis (detection wavelength 258 nm) of the xanthone enriched fraction from the ethanolic rhizome extract of *Anemarrhena asphodeloides*, the peaks represent: 1—neomangiferin, 2—mangiferin, 3—isomangiferin.

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
