# Peer review of "Anemarrhenae asphodeloides rhizoma Extract Enriched in Mangiferin Protects PC12 Cells against a Neurotoxic Agent-3-Nitropropionic Acid"

_ijms, 2020, doi:10.3390/ijms21072510_

Round 1
Reviewer 1 Report
The paper "Anemarrhenae rhizoma extract enriched in mangiferin protects PC12 cells against a neurotoxic agent - 3-nitropropionic acid" submitted to IJMS for publication describes an interesting use of an extract for neuroprotection.
The indroduction is pertinent with the data reported in the manuscript, introducing the reader to a full comprehension of the study.
Materials and methods are sufficiently pertinent and correct.
Please revise "ml" in all the manuscript. The correct form is "mL".
Please insert the chemical structures of the compounds reported in HPLC spectrum. It is a chemical-based work, so the reader has to perform SAR studies by reading.
One questions is:
Why did you use only one cell line, it would be interesting in the future evaluate these effects in other cell lines, rather than in vivo.
Author Response
Please, see the attachment (PDF)

Reviewer 2 Report
In this manuscript, the authors investigated the protective action of xanthones-enriched extract from A. asphodeloides rhizomes on the PC12 cell exposed to the neurotoxic agent - 3-nitropropionic acid (3-NP). In my opinion, data are still preliminary and lack of novelty. Several papers investigated the effects of plant nutrients/phytochemicals against the neurotoxic effects of 3-NP. A further molecular investigation is necessary to increase the scientific interest of this manuscript.
Other comments:
- Ref [5,6] should be updated
- RPMI 1640. Please indicate the product code
- serum (FBS; Sigma, USA??). ??
- Paragraph 3.5 is not clear. ....by fragmentation of the nucleus (weak Mitotracker Deep Red).... In my experience, Mitotracker stains mitochondria and not nuclei. A ratio Mitotracker green/red is necessary to better characterize the effects of these molecules on mitochondria number and function.
- Figure 4 and 5 size should be increased.
Author Response
Please, see the attachment (PDF)

Reviewer 3 Report
The research problem undertaken at work is interesting, however, in the manuscript, there are many places that must be revised
- L15-17 this sentence should be deleted
- L 36, add a reference
- Clarify the link between ROS and CNS, add an appropriate reference
- L39-41: The authors describe general and not specific works. please make a good summary of the relative and recent works which well matching with this study
- In this part, more recent references should be introduced (2017-2020)
- L50, what is the normally of 3-NP concentration of produced by A. flavus, add a suitable reference
- L54, clarify why 3-NP acted on biochemical and morphological changes in cells, describe carefully this part
- L157 On what basis authors used 5, 10 and 15 μg/ml of EAA.
- L67-70 loon sentence, make a good summary of this part
- Describe briefly the useful and active part of the plant ‘Anemarrhenae asphodeloides rhizome
- lack of references and discussion of Anemarrhenae asphodeloides rhizome activities: the authors must highlight this part (e.g recent works describing studied activity)
- L99, why authors studied the ethanolic extracts,
- L106-107, after 2 days of maceration by hexane, a high of losing active compounds specially phenolic compounds which are sensible of oxygen exposition, Here, authors should perform yield (recovery of the studied compounds before and after lmaceration (after 2 days)
- L 136, check this’ (FBS; Sigma, USA??),
- Section 2.2. Cell culture add suitable references for the studied methods, same remark for 2.3, 2.4 and 2.5 sections
- L16-199, delete this sentence,
- In figure 1 (A and B) authors should introduce the corresponding superscripts in each point (for each time and each concentration) to display the significance. Here, authors should use an appropriate statistical tool
- L 229 ‘3-NP such as 10 and 15 mM significantly increased mitoch’ when significantly was used, authors should introduce the statistical analysis (rewrite this part and use statistics to compare means
- L252-256 rewrite this sentence
- When you write significance, The significant difference, the significance of acclimation o please indicate that P< 0.05 or 0.01, or…
- L266-276 ‘PC12 cells were treated with various concentrations of EAA (0.5, 1, 5, 10, 15 and 20 μg/ml), 267 interchangeably with 3-NP (2.5, 5, 10, 15 mM), for 24h’ : A correlation study is needed in this study to interpret all the results (Pearson, Spearman, linear regression, …)
- Rewrite the 4. Discussion and make more clearer discussion with an appropriate correlation and appropriate statistical analysis: Really difficult to follow and understand the results. Can you improved the quality of all parts
- Article requires language corrections due to numerous errors
Author Response
Please, see the attachment (PDF)

Round 2
Reviewer 2 Report
The manuscript has been improved and major comments now addressed. I believe it is suitable for publication.
Reviewer 3 Report
accept